# Risk-adjusted Training and Evaluation for Medical Object Detection in Breast Cancer MRI

Dimitrios Bounias [1 2]   Michael Baumgartner [1 3 4]   Peter Neher [1 5 6]   Balint Kovacs [1 2]   Ralf Floca [1 7]
Paul F. Jaeger [4 8]   Lorenz A. Kapsner [9]   Jessica Eberle [9]   Dominique Hadler [9]   Frederik Laun [9]
Sabine Ohlmeyer [9]   Klaus H. Maier-Hein [1 2 3 5 6 10]   Sebastian Bickelhaupt [9]

## Abstract

Medical object detection revolves around discovering and rating lesions and other objects, with the most common way of measuring performance being FROC (Free-response Receiver Operating Characteristic), which calculates sensitivity at predefined thresholds of false positives per case. However, in a diagnosis or screening setting not all lesions are equally important, because small indeterminate lesions have limited clinical significance, while failing to detect and correctly classify high risk lesions can potentially hinder clinical prognosis and treatment options. It is therefore cardinal to correctly account for this risk imbalance in the way machine learning models are developed and evaluated. In this work, we propose risk-adjusted FROC (raFROC), an adaptation of FROC that constitutes a first step on reflecting the underlying clinical need more accurately. Experiments on two different breast cancer datasets with a total of 1535 lesions in 1735 subjects showcase

the clinical relevance of the proposed metric and its advantages over traditional evaluation methods. Additionally, by utilizing a risk-adjusted adaptation of focal loss (raFocal) we are able to improve the raFROC results and patient-level performance of nnDetection, a state-of-the-art medical object detection framework, at no expense of the regular FROC.

## 1. Introduction

In automatic diagnosis systems, discovery and rating of suspicious lesions in MR images is commonly formulated as an object detection task (Ayatollahi et al., 2021; Maicas et al., 2019; 2017; Gubern-Mérida et al., 2015). FROC (Free-response Receiver Operating Characteristic), which is calculated as the mean of sensitivity at predefined numbers of false positives per case, is the standard measure of performance (Reinke et al., 2022; Setio et al., 2017; Ayatollahi et al., 2021; Maicas et al., 2019; Baumgartner et al., 2021).

FROC and other detection metrics, as well as the losses used to train machine learning (ML) models, treat all objects as equally important. However, small indeterminate lesions often have lower clinical significance, while failing to detect and correctly classify high risk lesions can potentially hinder clinical prognosis and treatment options (Sopik & Narod, 2018). Additionally, if lower risk lesions are more prevalent the model can become biased towards them during training. Therefore, there is a disconnect between the underlying clinical need and the way models are optimized and evaluated, which can hinder clinical performance, make measuring scientific progress harder, and hamper clinical translation of ML (Maier-Hein et al., 2022).

Tumor size is one of the indicators of clinical prognosis in Breast Cancer (BC) and constitutes one of the three factors of the TNM (Tumor, Nodes, Metastasis) staging system (Cserni et al., 2018), which is widely used to categorize breast tumors and other carcinomas. Accordingly, the relationship between tumor size and mortality risk has been extensively studied (Sopik & Narod, 2018; Verschraegen

---
[1]Division of Medical Image Computing (MIC), German Cancer Research Center (DKFZ), Heidelberg, Germany [2]Medical Faculty, University of Heidelberg, Heidelberg, Germany [3]Faculty of Mathematics and Computer Science, University of Heidelberg, Heidelberg, Germany [4]Helmholtz Imaging, German Cancer Research Center (DKFZ), Heidelberg, Germany [5]German Cancer Consortium (DKTK), Partner Site Heidelberg, Germany [6]Pattern Analysis and Learning Group, Department of Radiation Oncology, Heidelberg University Hospital, Heidelberg, Germany [7]Heidelberg Institute of Radiation Oncology (HIRO), National Center for Radiation Research in Oncology (NCRO), Heidelberg, Germany [8]Interactive Machine Learning Group, German Cancer Research Center (DKFZ), Heidelberg, Germany [9]Institute of Radiology, Uniklinikum Erlangen, Friedrich-Alexander-Universität Erlangen-Nürnberg (FAU), Germany [10]National Center for Tumor Diseases (NCT), Heidelberg University Hospital (UKHD) and German Cancer Research Center (DKFZ), Heidelberg, Germany. Correspondence to: Dimitrios Bounias <dimitrios.bounias@dkfz-heidelberg.de>.

*Workshop on Interpretable ML in Healthcare at International Conference on Machine Learning (ICML)*, Honolulu, Hawaii, USA. 2023. Copyright 2023 by the author(s).

et al.), enabling the usage of tumor size as a characteristic of clinical significance for tumors and model predictions.

There are examples of methods that have taken similar concerns into account. Net benefit (Vickers et al., 2016) is an approach for evaluating classification models that sets an exchange rate between finding positive cases and performing unneeded biopsies. Also, in object detection, the non-medical COCO challenge (Lin et al., 2014) presents average precision results for small, medium, and large objects, alongside the total overall score. However, when applied to medical problems this size-stratified analysis suffers from various shortcomings. It requires selection of size thresholds, assumes that risk is uniform in each size range, utilizes different prediction probability thresholds for determining the number of false positives in each range, and can not result in a single score which can be compared. There is a need for more accurate and straightforward accounting of risk in medical object detection evaluation.

In this work, we present an adapted version of FROC, named *raFROC* ("risk-adjusted" FROC), that accounts for the risk differences between objects in medical object detection, bringing the evaluation metric closer to the needs of diagnosis and screening. Additionally, we analyze the performance of nnDetection (Baumgartner et al., 2021), a state-of-the-art medical object detection method built on Retina U-Net (Jaeger et al., 2020), using raFROC and compare it to other pre-existing metrics, in two BC datasets with different lesion size distributions and acquisition protocols, totaling 1535 lesions in 1735 subjects. Lastly, by utilizing a basic risk-adjusted adaptation of focal loss (*raFocal*), we are able to show that it is possible to get improvements in the raFROC results and patient-level performance, at no expense of the regular FROC.

## 2. Methodology

### 2.1. Risk-adjusted FROC (raFROC)

The FROC plot shows sensitivity at certain manually defined thresholds of false positives per case (commonly FPPI - False Positives Per Image) and constitutes the base for raFROC. To account for risk in raFROC, true positive (TP) predictions and ground truth samples are weighted by a weight $w \in [0, 1]$ pertaining to the associated risk, thus resulting in a risk-adjusted sensitivity. False positive (FP) predictions are in turn weighted by $(1-w)$, due to the desire to minimize unneeded biopsies and the lower value of low risk lesions, resulting in a risk-adjusted number of FPs (see Fig. 1). A TP prediction that has double the risk of another one will be given double the weight during evaluation, or vice versa for a FP. Since FROC is understood as sensitivity at manually defined thresholds of FPs per case, raFROC can be understood as approximating high-risk object sensitivity

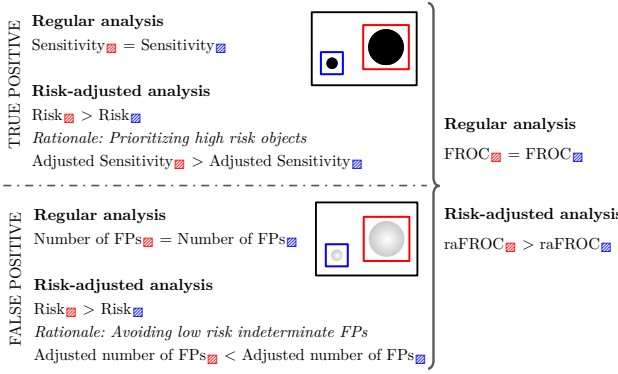

*Figure 1.* Example of the different weighting of objects in the calculation of the raFROC (risk-adjusted Free-response Receiver Operating Characteristic) metric. Red and blue represent different models.

at manually defined thresholds of low-risk FPs per case. Similarly to FROC, the final raFROC score is calculated by averaging the weighted sensitivities at all pre-defined FPPI thresholds (Dalmış et al., 2018; Niemeijer et al., 2011; Setio et al., 2017).

### 2.2. Risk-adjusted focal loss function (raFocal)

Focal loss (Lin et al., 2020) is a widely used loss function for training one stage detectors in the natural image processing domain. We propose a corresponding loss where focal loss is adapted to account for risk (raFocal). To that end, we asymmetrically adjust the loss for a *risk-adapted* class, a class for which the risk calculation applies. Detection often uses sigmoid binary classification and thus each class can be adapted independently. Loss for other potential foreground classes is left untouched and the methodology below only applies to the risk-adapted class.

Let $y \in \{\pm 1\}$ indicate sufficient IoU (Intersection-over-Union) with a ground truth object (with y$=-1$ indicating the background class). The risk-adjusted weight $w_t \in [1, 2]$ of a prediction is then calculated by:

$$w_t = \begin{cases} 1 + w_{gt}, & \text{if } y = 1 \\ 2 - w_{pred}, & \text{if } y = -1 \end{cases}$$

where $w_{gt} \in [0, 1]$ is the risk of the matching ground truth box and $w_{pred} \in [0, 1]$ is the risk of the prediction box.

Thus, if the prediction is assigned to a ground truth box, the loss gets weighted by a value that increases as the risk of the ground truth box becomes larger. Conversely, for a prediction assigned to the background class, the value

increases as the risk calculated for the prediction decreases.

The calculation for raFocal loss for the risk-adapted class finally becomes:

$$raFocal(p_t) \;=\; -w_t\alpha(1-p_t)^\gamma log(p_t)$$

where $p_t \in [0,1]$ is the model's estimated probability for the ground-truth class, $\alpha$ and $\gamma$ are the focal loss hyperparameters.

### 2.3. Risk estimation

In order to calculate risk for a lesion we incorporate the following function. It was calculated to be the 3rd degree polynomial fit of reported 15-year BC mortality based on lesion size in millimeters, in a multi-year study involving 819,647 patients (Sopik & Narod, 2018), which was the largest study available in an effort to minimize potential biases.

$$risk(size) \;=\; 2.28 \cdot 10^{-7} \cdot size^3 - 8.75 \cdot 10^{-5} \cdot size^2$$
$$+ \; 1.23 \cdot 10^{-2} \cdot size + 1.37 \cdot 10^{-3}$$

Fig. 2 shows the risk function alongside the data distribution. For use in this work, risk is normalized to $[0,1]$ by dividing by 0.641, the highest value in the $[0,150]$ mm range of the study.

While this function pertains to BC, similar functions can be constructed for other domains, potentially including other risk factors. The function can also be refined in the future if more epidemiological data are released.

## 3. Experiments

### 3.1. Data

**University Hospital (UH) dataset.** The ethics committee of Friedrich-Alexander-University (FAU) Erlangen-Nürnberg approved this study and waived the need for written informed consent. Inclusion critiria were acquisitions with full diagnostic protocol that included DWI (Diffusion Weighting Imaging) sequences acquired with the b-values b=50, 750, 1500 $s/mm^2$, using 1.5T and 3.0T MRI. The dataset comprises 818 patients and included examinations were acquired between November 2017 and January 2020. Lesions were segmented in DWI sequences as a consensus (one board certified radiologist supervising one medical student). Histopathologically proven malignant lesions account for 618 lesions in 268 cases, while 1003 lesions in 373 cases were kept as a benign class to aid the model.

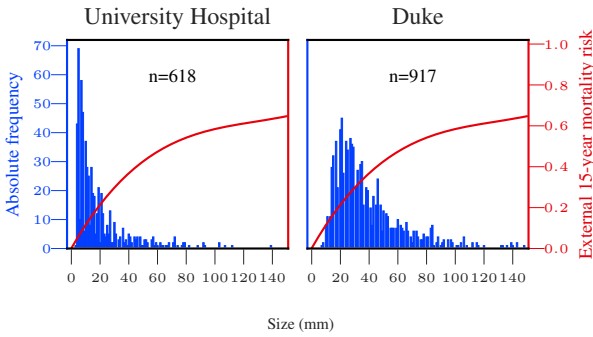

**Data distribution**

*Figure 2.* Absolute frequencies of malignant tumor sizes, alongside the associated 15-year mortality risk of each size. The University Hospital dataset is DW-MRI and Duke is DCE-MRI. Size is defined as the max axial detection box side.

**Duke dataset.** Publicly available (Saha et al., 2022; 2018; Clark et al., 2013), pre-operative dynamic contrast enhanced MRI, including 917 cases with 917 lesions (randomly selected independent test set: 230 images / 230 lesions). Subtraction images were used, in which the pre-contrast image was subtracted from the second post-contrast image.

### 3.2. Experimental setup

nnDetection was used as the development framework. Model parameters were optimized solely using focal loss and FROC. 8 epochs were found to perform best with 2500 batches per epoch and a learning rate of 0.01. The standard nnDetection augmentation pipeline was used. Focal loss hyperparameters $\alpha$=0.75 and $\gamma$=1 were used (from values in $[0.25, 0.75]$ and $\{1, 2\}$ respectively). In order to assess the validity of the proposed loss and metric, we perform the following experiments:

**Empirically confirm shortcomings of FROC to motivate raFROC.** We assume two hypothetical models, where one is equivalent to detecting only small lesions and another to detecting only large ones. To achieve that, we pick a size threshold that splits the lesions into two buckets as evenly as possible. The two hypothetical models are able to predict lesions only from their respective bucket. We investigate whether there is a difference in patient-level sensitivity and risk-adjusted object-level sensitivity. This experiment is performed in the UH dataset, which contains both pathological and non-pathological cases.

**Performance of proposed loss using traditional evaluation metrics.** We evaluate by means of aggregating 5-fold cross validation results for the UH dataset and dedicated test set for the Duke dataset. The methods used are:

- **FROC** analysis for all datasets.

- **Size-stratified FROC** analysis for all datasets using the above-mentioned COCO method (Lin et al., 2014), where for a certain object size range, ground truth objects with size outside the range as well as non matching predictions with size outside the range are discarded prior to the calculation. 20mm was chosen as the size range cut-off, because it constitutes the cut-off between T1 and T2 in the TNM staging system (Cserni et al., 2018).

- **Patient-level AUC** (Area Under Curve) and **patient-level AP** (Average Precision) for the UH dataset that has both pathological and non-pathological cases.

**Performance of proposed loss using the proposed evaluation metric.** The hypothesis is that raFocal will perform better than focal loss, especially for the UH dataset where there is an abundance of small lesions. We are also looking to gain insights on the risk-adjusted performance using the evaluation performed with the traditional methods and showcase that raFROC is a simpler approach with less overhead and pitfalls.

### 3.3. Results

**Empirically confirm shortcomings of FROC to motivate raFROC.** We chose a size threshold of 12mm, in order to split the ground truth malignant lesions into two buckets as evenly as possible, which resulted in 302 lesions in the small lesion size bucket and 316 in the large one. We assume two hypothetical models, where the first one detects all of the small lesions and the second all the large ones. These models have similar object-level sensitivities, 0.49 and 0.51, as they are able to detect a similar amount of malignant lesions. However, the diagnostic value of the two models differs, as smaller lesions are often accompanied by larger ones in the same patient and because there can be multiple small lesions per patient. The small lesion model can achieve a patient-level sensitivity of only 0.55, compared to 0.82 for the large lesion one. As such, the large lesion model would be preferable in a diagnostic setting. The risk-adjusted object-level sensitivity that raFROC uses is able to reflect this difference by using increased weights for larger lesions, coming out as 0.24 and 0.76 respectively. Fig. 3 summarizes the result.

**Performance of proposed loss using traditional evaluation metrics.** Fig. 4 shows the results of the FROC analysis.

- For the **UH** dataset, the raFocal model performs slightly better at the object-level in lower thresholds. The higher sensitivity of the focal model at 1 FP/case in the 20+mm plot is misleading, as this happens for a very low prediction probability threshold (0.025) which

**Higher patient-level predictive ability of large high-risk lesions**

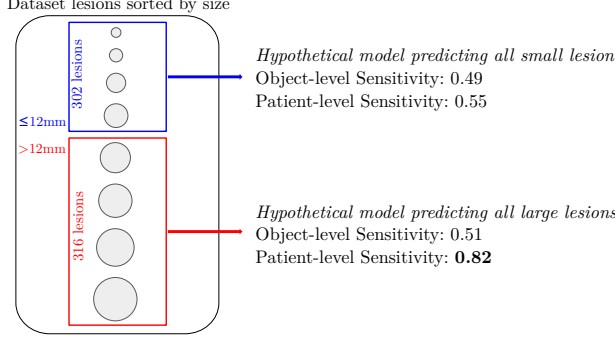

Dataset lesions sorted by size

*Hypothetical model predicting all small lesions*
Object-level Sensitivity: 0.49
Patient-level Sensitivity: 0.55

*Hypothetical model predicting all large lesions*
Object-level Sensitivity: 0.51
Patient-level Sensitivity: **0.82**

*Figure 3.* Summary of the different predictive ability of lesion sizes in the experiment on the University Hospital data, where all lesions are divided into two buckets depending on their size and two hypothetical models are assumed that predict only from the respective bucket. Multiple lesions can belong to a single patient.

is not relevant, as it is lower than the largest threshold used in the overall FROC plot at 8 FPs/case (0.039). On the patient level, the raFocal model achieves 0.86 AUC and 0.77 AP compared to 0.84 AUC and 0.70 AP for the focal model (p=0.006 significant difference between AUC scores using the DeLong method (Sun & Xu, 2014)), meaning that the raFocal model was more capable of detecting pathological cases.

- For the **Duke** dataset, the analysis shows improvement in all FROC plots. However, upon closer inspection, most of the benefit in the overall FROC performance comes from the 20+mm range and the two plots follow a similar pattern. That is because the probability thresholds used for the overall FROC curve produce very similar number of 0-20mm TPs. In fact, the largest difference is in the lowest threshold, where focal loss actually has more 0-20mm TPs, 24 vs 20, something not visible in the 0-20mm plot.

**Performance of proposed loss using the proposed evaluation metric.** The results (Fig. 5) indicate that raFocal shows improvements in raFROC compared to focal loss, especially for the UH dataset, where there is more variance in the lesion sizes. This is in accordance with the increase in the patient-level metrics observed and the slight increase in 0-20mm lesion performance for the lower thresholds. The improvement in the Duke dataset stems from better large lesion performance, as indicated by the size-stratified analysis. raFROC is able to incorporate components from all pre-existing evaluation techniques, while keeping results simple and reflecting a more accurate risk model, regardless of the underlying properties of the dataset.

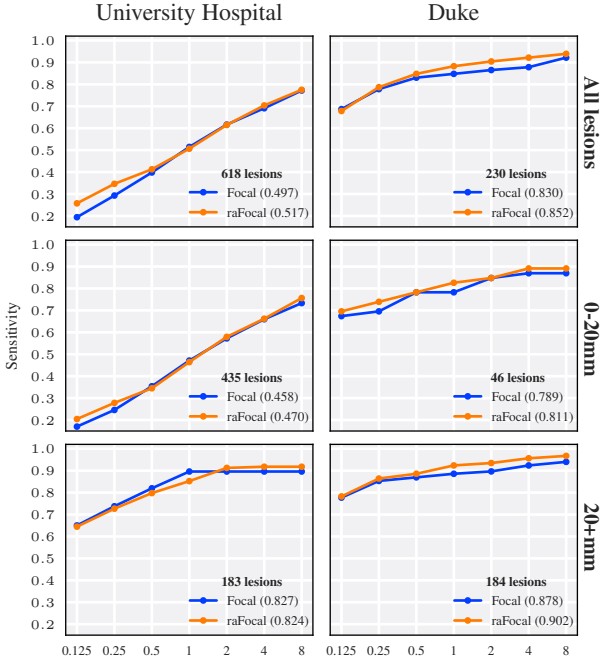

*Figure 4.* Performance of the proposed loss function (raFocal) using traditional evaluation metrics, in a DW-MRI (University Hospital) and a DCE-MRI (Duke) breast cancer datasets. The first row is the regular FROC (Free-response Receiver Operating Characteristic), while the next two rows comprise the size-stratified FROC analysis with the method used in COCO.

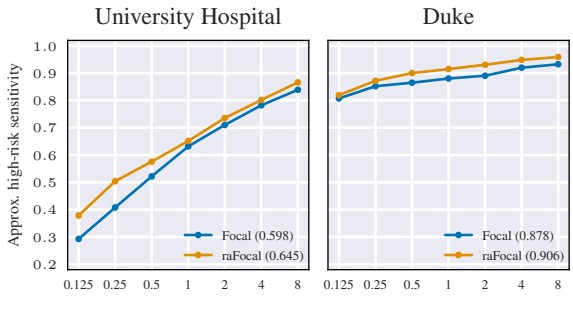

*Figure 5.* Performance of the proposed loss function (raFocal) using the proposed evaluation metric, raFROC (risk-adjusted Free-response Receiver Operating Characteristic).

## 3.4. Discussion

In this work, we propose a risk-adjusted approach to training and evaluating medical object detection models using DW and DCE MRI breast cancer datasets.

While FROC is a useful metric for evaluation in medical object detection, it can only speak to a model's ability in detecting objects, without any other consideration regarding the dataset distribution, patient-level performance, and risk disparities between objects. Size-stratified FROC, as used in COCO (Lin et al., 2014), can potentially showcase differences in model performance for different object size ranges, but it comes with the limitations discussed above and can sometimes even be misleading by showing results of prediction probability thresholds that are not applicable. Separate probability thresholds for each range are not possible, as the method allows for predictions to be counted as correct if matching, regardless of size. Patient-level metrics can be of similar importance. However, they add new parameters to the decision process, making it difficult to decide between two potential models if, e.g., one of them has higher FROC and the other higher AUC. Risk-adjusted FROC is able to incorporate aspects of all aforementioned metrics and is situated closer to the underlying clinical need, as it is augmented by well established adverse clinical outcome data and can pave the way for incorporating more risk and clinical pathway considerations into medical machine learning.

Adequate loss functions are required to represent the addressed problem and data, thereby accounting for any imbalances. If a dataset has many small lesions, the model can be lead to try capture more of them at the expense of larger ones. The proposed raFocal loss constitutes a first example methodology for bringing risk considerations into the loss calculation for BC analysis. Additionally, given that having multiple high risk large lesions in a single case is quite common, raFocal can reflect patient-level performance alongside object-level performance, which was previously missing during model training.

Utilizing only lesion size is one of the limitations of this study; it is however a first step in the right direction. Size is one of the important factors influencing prognosis and there is currently a lack of additional well-established risk factors provided by MR imaging alone. The moderate performance of the neural network in the UH dataset could be seen as another limitation of our work. It can however be explained by the large amount of 1003 small benign lesions, often present in cases where there are also malignant lesions, in non contrast-enhanced MRI.

# 4. Conclusion

This work showcases how to realize risk-adjusted model training and validation in medical object detection. Accounting for clinical risk and outcome is very important in a medical diagnosis setting compared to other domains, because it allows balancing the trade off between false positive findings and missing pathologies, significantly influencing the clinical outcome of the individual patient. The method presented brings model evaluation and training loss closer to that need and is a better approach than size stratification. Source code for raFROC and raFocal loss is publicly available on https://github.com/MIC-DKFZ/imlh-icml-detection-tools.

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

## A. Patient-level Predictive Ability of Lesion Sizes Results Summary

*Table 1.* Summary of experiment: Empirically confirm shortcomings of FROC to motivate raFROC.

|  | NUMBER OF LESIONS | OBJECT SENS | PATIENT SENS | RISK-ADJUSTED SENS |
|---|---|---|---|---|
| SMALL LESION BUCKET | 302 | 0.49 | 0.55 | 0.24 |
| LARGE LESION BUCKET | 316 | 0.51 | 0.82 | 0.76 |

## B. Loss Comparison Results Summary

*Table 2.* Summary of focal and raFocal loss results in the two datasets.

|  | UNIVERSITY HOSPITAL | | | | DUKE | |
|---|---|---|---|---|---|---|
| LOSS | AUC | AP | FROC | RAFROC | FROC | RAFROC |
| FOCAL | 0.84 | 0.70 | 0.497 | 0.598 | 0.830 | 0.878 |
| RAFOCAL | 0.86 | 0.77 | 0.517 | 0.645 | 0.852 | 0.906 |

