# OpenReview forum: "Risk-adjusted Training and Evaluation for Medical Object Detection in Breast Cancer MRI"
_ICML.cc/2023/Workshop/IMLH — IMLH 2023 Poster_

### Official Review · Reviewer_y4p7 · 2023-06-16
**risk-adjusted FROC and risk-adjusted adaptation of focal loss**

**Rating:** 7
**Confidence:** 4

**Review:**

Summary and Strength:
The paper addresses the issue of correctly accounting for the risk imbalance in medical object detection, where not all lesions are equally important. The authors propose risk-adjusted FROC (raFROC). The paper presents experiments on two different breast cancer datasets with a total of 1535 lesions in 1735 subjects, which showcase the clinical relevance of the proposed metric and its advantages over traditional evaluation methods. Additionally, the authors use a risk-adjusted adaptation of focal loss (raFocal) to improve the raFROC results and patient-level performance without compromising traditional FROC performance. Overall, the results suggest that raFROC and raFocal can better account for the risk imbalance in medical object detection and improve the clinical relevance of the evaluation and performance metrics.

Weakness:
- The network used in experiment is absent. It is recommended that the authors report the network architecture used to provide readers with a better understanding of the experimental setup.
- More networks are encouraged to conduct and validate the effectiveness of proposed loss and evaluation metrics. Conducting additional experiments on a wider range of network can help establish the robustness and generalizability of the proposed methods.

---

### Official Review · Reviewer_Ap6D · 2023-06-17
**Well-written paper with promising results**

**Rating:** 7
**Confidence:** 2

**Review:**

This work describes how to realise risk-adjusted model training and validation in medical object detection. The authors present an adapted version of FROC, named raFROC (“risk-adjusted” FROC), that accounts for the risk differences between objects in medical object detection, bringing the evaluation metric closer to the needs of diagnosis and screening.

The paper is overall well-written and organised. The quality of experiments are good and the results look promising. I have no major concerns.

---

### Official Review · Reviewer_PSbh · 2023-06-18
**Further investigation on other risk factors is needed.**

**Rating:** 5
**Confidence:** 5

**Review:**

This article improves the detectability and diagnostic value of medical subject testing through a risk-adjusted FROC (raFROC) approach. Unlike traditional assessments, raFROC considers inter-subject risk differences and more accurately reflects clinical needs. In addition to this, the article proposes a risk-adjusted loss-of-focus, raFocal, which improves raFROC results and patient-level performance. The experimental results show that raFROC and raFocal can more accurately and simply calculate risk in the assessment of medical subject testing, and can improve detection power and diagnostic value.
This study has application value in proposing better assessment metrics that can be better adapted to clinical needs for breast cancer data. However, the analysis was performed only for lesion size as a risk factor, further investigation on other risk factors is needed. And more experimental comparisons of indicators are highly recommended.

---

### Official Review · Reviewer_vJnq · 2023-06-19
**Review for raFROC paper**

**Rating:** 7
**Confidence:** 4

**Review:**

__Overview__: The authors propose raFROC and raFocal, which are methods for adjusting FROC and focal loss to align with perceived risk (i.e. mortality) of medical objects. In particular, breast cancer tumor detection datasets are biased towards small objects but large objects have a higher associated risk.

__Positives__:
1. Correctly evaluating methods, such that they align with the goals of the real world downstream tasks, is a critically important topic. As this paper demonstrates, the metrics typically used for evaluating object detection methods do not correctly capture important information necessary for breast cancer screening. The authors include a strategy for both evaluation and training models which captures risk.
2. Well written paper that was easy to follow.

__Improvements__:
1. Unclear how raFROC differs from previously established weighting strategies for ROC, such as in the following paper:
> _Maurer, Andreas, and Massimiliano Pontil. "Estimating weighted areas under the ROC curve." Advances in Neural Information Processing Systems 33 (2020): 7733-7742._
2. Would benefit from investigating different types of risk correlates or different types of pathologies.
3. The method is a bit simple.

---

### Meta-Review · Area_Chair_MCXS · 2023-06-20

**Recommendation:** Accept (Poster)
**Confidence:** 5

**Metareview:**

The authors discussed a crucial topic on training strategies for medical object detection. Controlling risk in a complicated machine learning system for diagnosis is challenging. The paper received positive feedback from most reviews, indicating that it meets the high standards required for presentation to the community. I suggest the authors can further discuss the concerns from reviewers regarding: 1) highlight the novelty of the proposed method; 2) justify the criteria of metric selections for risk facto, e.g., lesion size only. 3) Network selection. Overall, this paper is in good shape, and present good studies to the explainable AI in healthcare.

---

### Decision · Program_Chairs · 2023-06-20

Accept (Poster)